# Efficient and Effective Multi-task Grouping via Meta Learning on Task Combinations

**Xiaozhuang Song**[*]
Southern University of Science and Technology
shawnsxz97@gmail.com

**Shun Zheng**
Microsoft Research
shun.zheng@microsoft.com

**Wei Cao**
Microsoft Research
wei.cao@microsoft.com

**James J.Q. Yu**
Southern University of Science and Technology
yujq3@sustech.edu.cn

**Jiang Bian**
Microsoft Research
jiang.bian@microsoft.com

## Abstract

As a longstanding learning paradigm, multi-task learning has been widely applied into a variety of machine learning applications. Nonetheless, identifying which tasks should be learned together is still a challenging fundamental problem because the possible task combinations grow exponentially with the number of tasks, and existing solutions heavily relying on heuristics may probably lead to ineffective groupings with severe performance degradation. To bridge this gap, we develop a systematic multi-task grouping framework with a new meta-learning problem on task combinations, which is to predict the per-task performance gains of multi-task learning over single-task learning for any combination. Our underlying assumption is that no matter how large the space of task combinations is, the relationships between task combinations and performance gains lie in some low-dimensional manifolds and thus can be learnable. Accordingly, we develop a neural meta learner, MTG-Net, to capture these relationships, and design an active learning strategy to progressively select meta-training samples. In this way, even with limited meta samples, MTG-Net holds the potential to produce reasonable gain estimations on arbitrary task combinations. Extensive experiments on diversified multi-task scenarios demonstrate the efficiency and effectiveness of our method. Specifically, in a large-scale evaluation with 27 tasks, which produce over one hundred million task combinations, our method almost doubles the performance obtained by the existing best solution given roughly the same computational cost. Data and code are available at `https://github.com/ShawnKS/MTG-Net`.

## 1 Introduction

Multi-task learning (MTL), as a longstanding learning paradigm [37, 50], has been widely applied into a variety of machine learning applications, ranging from language understanding [11], visual recognition [33, 41], and robotic control [19] to drug discovery [36] and clinical therapeutics [17, 30]. The major motivation behind is to boost the performance of single-task learning (STL) by leveraging the additional supervision signals from other relevant tasks.

---

[*]Work is done during the internship at Microsoft Research Asia.

36th Conference on Neural Information Processing Systems (NeurIPS 2022).

Nevertheless, naively grouping multiple tasks together for MTL easily results in severe performance degradation in practice. Many recent studies acknowledged this *negative-transfer* phenomenon and speculated that the major reason lied in the competition and interference among incompatible tasks [39, 47, 15, 41, 14, 30, 2, 6]. To mitigate such negative-transfer effect, recent years have witnessed a few explorations on developing new optimization approaches [39, 25, 29, 48] or model architectures [27, 33, 28, 15]. But explicitly identifying a group of tasks that would benefit from training together still remains a challenging fundamental problem without adequate investigations [41, 14]. Thoroughly studying this *multi-task grouping* (MTG) problem not only advances core research over MTL aside from optimization or architecture developments but also facilitates various MTL scenarios that can benefit from learning auxiliary objectives [2, 6].

However, identifying the best MTG solution is extremely challenging as it involves an exhaustive search over the whole space of task combinations, each of which corresponds to a whole procedure of model training and evaluation on its specific MTL objective. The cost of such an exhaustive search grows exponentially with the number of tasks and thus can be prohibitively expensive [6, 41, 14].

To reduce this prohibitive computational cost, most recent efforts [41, 14] relied on the assumption of high-order approximations (HOA), meaning *high-order performance gains can be estimated from corresponding pairwise gains by averaging*, where the performance gain denotes the improvement of MTL over STL for a specific task. For instance, considering the group composed of tasks {A, B, C}, they assumed that when learning with the joint objective of this group, the performance gain for A can be estimated by the average of associated pairwise gains for A, which were independently obtained by learning with {A, B} and {A, C}. Under the HOA assumption, they only need to collect the transferring gains for all pairwise combinations, of which the number is only quadratically dependent on the number of tasks. While substantially reducing computational requirements, the HOA assumption also results in wildly inaccurate estimations when there exist non-linear relationships between high-order gains and corresponding pairwise gains. As a result, such untrustworthy estimations can significantly hinder the final performance of searching for the optimal grouping option.

To pursue more accurate estimations of various high-order performance gains under affordable computational cost, we develop a systematic MTG framework with a new meta-learning problem on task combinations. To be specific, we divide all task combinations into two parts: one with a small number of combinations for meta training and the other with all the rest combinations for meta testing. Here the meta input is a specific task combination, and the meta objective is to predict the per-task performance gains (the meta label) for this combination. The underlying assumption of this meta-learning formulation is that *although the number of task combinations grow exponentially with the number of tasks, the relationships between task combinations and performance gains lie in some low-dimensional manifolds and thus can be learnable*. Following this formulation, we build a dedicated neural network, MTG-Net, as the meta learner, and further develop an active learning [35] strategy to progressively construct the meta-training set. With this strategy, we can largely improve the effectiveness of MTG-Net with only a small number of meta-training samples. After the meta-training stage, we leverage the gain predictions of MTG-Net on all task combinations to guide the final grouping selection. Extensive experiments on diversified multi-task scenarios, including vision, energy, and healthcare, demonstrate the efficiency and the effectiveness of our method. Moreover, we also visualize various latent structures discovered by MTG-Net, which empirically validate the practicability of our assumption.

In summary, our contributions include:

- To our best knowledge, this paper is the first effort to formulate a meta-learning problem for MTG and thus enable an efficient exploration of the exponentially growing space of task combinations, which was believed to be prohibitively challenging [41, 30, 2, 6].

- We build a neural meta learner, MTG-Net, that can capture the relationships between task combinations and performance gains effectively.

- We develop an active learning strategy to progressively select the most useful meta-training samples for efficient training of MTG-Net.

- We conduct extensive experiments across diversified multi-task scenarios. Specifically, in a large-scale evaluation (27 tasks) with over one hundred million task combinations, our method almost doubles the performance obtained by the existing solution.

## 2  Related Work

**MTG**   The classic assumption of MTL in statistical learning was that putting related tasks into joint learning benefited from the inductive bias encouraging cross-task information sharing [8, 9, 7, 3, 5]. However, when this assumption of relatedness did not hold, the resulting *negative-transfer* effects can cause significant performance degradation. Thus the further research investigated the problem of simultaneously determining with which tasks each task should share while carrying over classical cross-task sharing [20, 22]. Nevertheless, these early MTL studies can hardly be adapted to modern deep neural networks [24] due to certain prohibitive assumptions [14]. As mentioned by [41, 14], identifying proper task groupings for deep neural networks conventionally required either computationally intensive cross-validation procedures or the human knowledge that is not always applicable to machine learning. The early attempt [41] on more general and systematic MTG for deep neural networks formalized a standard workflow: 1) collecting the transferring gains (or gain predictions) for all $2^N - 1$ task combinations, and then 2) conducting a brute-force search for the best grouping option given a specific budget, such as the maximal number of groups. The latter step is a deterministic searching procedure, but the former step is rather challenging because the optimal solution requires $2^N - 1$ times MTL training and evaluation, which can be prohibitively expensive when $N$ increases. Thus [41] further proposed the HOA approximation to reduce the computational complexity from exponential into quadratic: $\binom{N}{2}$ times MTL on pairwise task combinations. Subsequently, [14] improved the efficiency of [41] by substituting $\binom{N}{2}$ times pairwise MTL procedures with a single run, in which the computation complexity is only linearly dependent on $N$ due to maintaining and updating a task affinity matrix on the fly per each gradient updating. This further approximation, named as task affinity grouping (TAG) in [14], traded the accuracy in estimating pairwise MTL gains for efficiency, which may result in more error propagation from pairwise to higher-order. Different from all these HOA-based methods, this paper formulates a systematic meta-learning problem to estimate the transferring gains and builds an effective meta model to enable more accurate and robust generalization to massive high-order combinations.

**Other Research on MTL & Meta Learning & Task Embedding**   Recent years witnessed plenty of explorations on MTL from other aspects, such as multi-objective optimization [39, 25, 29, 48], neural architecture search [28, 26, 16, 15, 42], and soft-sharing mechanisms [13, 33, 27]. While these studies have made remarkable progresses to improve MTL under a predefined task combination, the focus of this paper lies in *explicitly determining with which tasks to share*. Besides, as pointed by [37, 14], these two paradigms are complementary to each other and can contribute to performance improvements jointly. Moreover, our work falls into the paradigm of meta learning [43, 21, 4, 34, 40]. But different from traditional meta-learning studies, which primarily focused on the fast adaptation across tasks in few-shot scenarios, our meta objective is to estimate the performance gains for different task combinations. To fulfill this objective, we develop a meta network that treats a task combination as a set of task tokens and transforms them into task embeddings to learn their interactions. Meanwhile, we also note some existing studies leveraged a similar idea of learning task embeddings but for different purposes, such as inferring task similarities [1], estimating treatment effects [38], facilitating few-shot learning [10] or reinforcement learning [23], etc.

## 3  Meta Learning for Multi-task Grouping

In this section, we elaborate on how our meta learning framework works for MTG. First, in Section 3.1, we establish the basic notations and briefly introduce some crucial steps of MTG for better understanding. Then we formulate our meta learning problem on task combinations in Section 3.2. Next, we introduce the details of MTG-Net and the accompanied active learning strategy in Sections 3.3 and 3.4, respectively. Figure 1 gives an overview of our framework.

### 3.1  Preliminaries

**Notations**   We adopt the following notations throughout this paper. $N$ is the number of learning tasks in total. $\mathcal{T} = \{T_1, T_2, \cdots, T_N\}$ is the universal set of all $N$ tasks, where $T_j$ denotes the $j$-th task, and we have $|\mathcal{T}| = N$. $\mathcal{C} = \{C_1, C_2, \cdots, C_{2^N-1}\}$ is the universal set of all $2^N - 1$ combinations, which contains $\binom{N}{1}$ combinations containing only one task, $\binom{N}{2}$ combinations containing exact two tasks, $\binom{N}{3}$ three-task combinations, etc. $C_i = \left[t_1^i, t_2^i, \cdots, t_{|C_i|}^i\right]$ denotes the $i$-th task combination, where

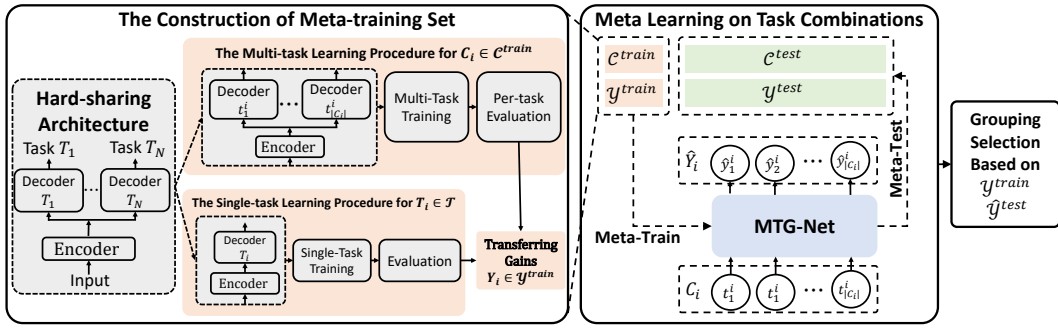

Figure 1: An overview of our meta learning framework for multi-task grouping

$t_j^i \in \mathcal{T}$ denotes the $j$-th task in this combination. $\mathcal{Y} = \{Y_1, Y_2, \cdots, Y_{2^N-1}\}$ is the universal set of all transferring gains corresponding to $\mathcal{C}$. $Y_i = [y_1^i, y_2^i, \cdots, y_{|C_i|}^i]$ includes all per-task transferring gains for task combination $C_i$, where $y_j^i$ denotes the gain of conducting MTL on task combination $C_i$ over performing STL for task $t_j^i$, and we have $|Y_i| = |C_i|$. $\mathcal{C}^{train}$ is the set of task combinations for meta training, and $\mathcal{Y}^{train}$ denotes the set of corresponding transferring gains. $\mathcal{C}^{test}$ is the set of task combinations for meta testing, and $\mathcal{Y}^{test}$ also includes corresponding transferring gains. $B$ denotes the budget in grouping selection, which is the maximal number of selected task combinations.

**The MTL Procedure**    Given task combination $C_i$, we define its MTL procedure as training with the joint objective of $C_i$ and evaluating on the validation set for each task in $C_i$ to collect per-task performance as an indication of generalization. In accordance with [41, 14], we adopt a widely used hard-sharing architecture for MTL, in which there is a single encoder to obtain shared representations followed by separate decoders for specific tasks. Moreover, we retain the same split of train, validation, and test sets for each MTL procedure and fix the optimization algorithm as well as other hyper-parameters. In this way, we can attribute the variations of generalization performance for a specific task largely to the different MTL objectives of various task combinations.

**Transferring Gain**    Here we make more clarifications on collecting transferring gain $y_j^i$ for task $t_j^i$ by performing MTL on combination $C_i$. First, we conduct the aforementioned MTL procedure to collect the generalization performance on the validation set for task $t_j^i$ in $C_i$. Then, we define the transferring gain of conducting MTL on $C_i$ for $t_j^i$ as the improvement of the performance of MTL over the base performance of STL for $t_j^i$. In practice, we can instantiate this improvement in different ways, such as the reduction of loss, the improvement in accuracy, etc.

**Grouping Selection**    Once obtaining all transferring gains, we can select the optimal groupings that not only possess the best overall performance but also meet some practical considerations, such as inference latency. This selection procedure is in essence a constrained searching problem, which is to select a group of task combinations covering all $N$ tasks to maximize the overall transferring gain (the average of per-task gain) while ensuring the number of groups does not exceed a given budget ($B$). As noted by [41, 14], this problem is NP-hard in general. A solution of recursive searching has the computational complexity of $O(2^{N \cdot B})$. In our experiments, when $N$ is small (e.g., 5 vision tasks), we follow [41, 14] to use the branch-and-bound-like algorithm to search for the optimal groupings. While for large $N$ and $B$ (e.g., 27 medical tasks), the exact search becomes prohibitively expensive. In this case, we approximately select near-optimal groupings via a beam-search approach with polynomial complexity, which is attached in the appendix.

### 3.2    Meta Learning on Task Combinations

Different from all existing brute-force or heuristic approaches, we formulate a new meta-learning problem on task combinations as a more systematic framework for the estimations of transferring gains. As mentioned in preliminaries, a predefined configuration of the MTL procedure implies that the variations in transferring gains ($Y_i$) can be largely attributed to the MTL objectives of different task combinations ($C_i$). Therefore, we assume there is an underlying mapping $\mathcal{F}$ to determine the

transferring gains given a specific task combination: $Y_i = \mathcal{F}(C_i)$, for $C_i \in \mathcal{C}$. Furthermore, this view stimulates us to design a learning process to obtain a good approximation $\mathcal{F}^*$ to the oracle function $\mathcal{F}$ with limited meta samples from $\mathcal{C}$. The underlying assumption of the learnability with limited meta samples is that the relationships between task combinations and performance gains lie in some low-dimensional manifolds no matter how large the space of task combinations is. In our experiments, we empirically validate the practicability of this assumption.

To be specific, we divide the whole set $\mathcal{C}$ into two parts: one part $\mathcal{C}^{train}$ for meta training and the other $\mathcal{C}^{test}$ for meta testing. For every combination in $\mathcal{C}^{train}$, we conduct an MTL procedure and collect corresponding transferring gains ($\mathcal{Y}^{train}$) to learn a surrogate function $\mathcal{F}^*$. While for all combinations in $\mathcal{C}^{test}$, we directly apply $\mathcal{F}^*$ to produce estimated transferring gains, the set of which is denoted as $\hat{\mathcal{Y}}^{test}$. Accordingly, we summarize the whole meta-learning procedure as

$$\mathcal{F}^* = \texttt{Meta-Train}\big(\mathcal{C}^{train}, \mathcal{Y}^{train}\big), \quad \hat{\mathcal{Y}}^{test} = \texttt{Meta-Test}\big(\mathcal{C}^{test}, \mathcal{F}^*\big). \tag{1}$$

This meta-learning perspective provides an explicit trade-off between efficiency and effectiveness for MTG. On the one hand, as the size of $\mathcal{C}^{train}$ corresponds to the major computational cost, decreasing $|\mathcal{C}^{train}|$ helps improving efficiency. On the other hand, since the estimation performance on $\mathcal{C}^{test}$ largely determines the quality of the final grouping decisions, increasing $|\mathcal{C}^{train}|$ fosters effectiveness. Besides, given a fixed budget of $|\mathcal{C}^{train}|$, namely affordable computational cost, we can devote more efforts to develop advanced meta-learning strategies to improve the gain estimation. Moreover, this new view can incorporate the well-known HOA assumption as a special case. The HOA assumption groups all pairwise task combinations into $\mathcal{C}^{train}$ and takes the average of pairwise gains as the approximation of high-order gains for all other combinations in $\mathcal{C}^{test}$. Thus we can view it as an nonparametric method to represent the oracle mapping $\mathcal{F}$.

### 3.3 MTG-Net

In the following, we elaborate on the design of our meta learner. Specifically, we call this meta learner $\texttt{MTG-Net}$, namely a meta network to facilitate multi-task grouping. First, to provide rich representations for discrete task indicators, we introduce a classic technique from natural language research [32]: transforming discrete work tokens into vector representations while preserving semantic relationships. In our case, we also construct an embedding table $\mathbf{E} \in \mathbb{R}^{N \times D}$ to transform the input $C_i$ into a vector of embeddings, denoted as $\mathbf{X}_i = \big[\boldsymbol{x}_1^i, \boldsymbol{x}_2^i, \cdots, \boldsymbol{x}_{|C_i|}^i\big] \in \mathbb{R}^{|C_i| x D}$, where $D$ denotes the embedding dimension, and $\boldsymbol{x}_j^i \in \mathbb{R}^{1 \times D}$ corresponds to the embedding for task $t_j^i$.

Afterwards, to effectively capture the diversified interactions in different task combinations, we stack several self-attention encoding layers [45] over the dense representation $\boldsymbol{X}_i$. The reason is that the self-attention mechanism not only achieves remarkable successes in a wide variety of applications [12, 18] but also perfectly matches the unordered property of task combination and holds sufficient capacities to model those diversified inter-task interactions. Here we summarize this encoding procedure as $\boldsymbol{H}_i = \texttt{Self-Att-Enc}\big(\boldsymbol{X}_i\big)$, where $\boldsymbol{H}_i \in \mathbb{R}^{N \times D}$ is the vector of enriched representations that encode diversified interactions among tasks, and $\texttt{Self-Att-Enc}$ is the self-attention-based encoder developed by [45]. Moreover, another benefit of $\texttt{Self-Att-Enc}$ is that the encoded vector $\boldsymbol{H}_i$ shares the same length with the label vector $Y_i$. Naturally, we stack a linear mapping over $\boldsymbol{H}_i$ to obtain the final output as $\hat{Y}_i = \boldsymbol{H}_i \cdot \boldsymbol{w} + b$, where the vector $\hat{Y}_i = \big[\hat{y}_1^i, \hat{y}_2^i, \cdots, \hat{y}_{|C_i|}^i\big] \in \mathbb{R}^{|C_i|}$ corresponds to our estimation of $Y_i$, $\hat{y}_j^i$ corresponds to the estimated gain for $t_j^i$, $\boldsymbol{w} \in \mathbb{R}^D$ is a weight parameter, and $b \in \mathbb{R}^1$ is a scalar parameter being broadcasted to each entry of $\boldsymbol{H}_i \cdot \boldsymbol{w}$.

In summary, MTG-Net takes in a task combination ($C_i$) and emits a collection of gain estimations ($\hat{Y}_i$). Recalling Equation (1), we formulate the specific meta-learning procedure for MTG-Net as:

$$\texttt{Meta-Train:} \quad \Theta^* = \arg\min_{\Theta} \sum_{C_i \in \mathcal{C}^{train}} \|\hat{Y}_i - Y_i\|_2^2,$$

$$\texttt{Meta-Test:} \quad \hat{\mathcal{Y}}^{test} = \big\{\texttt{MTG-Net}_{\Theta^*}(C_i), \text{ for } C_i \in \mathcal{C}^{test}\big\}, \tag{2}$$

where $\hat{Y}_i = \texttt{MTG-Net}_{\Theta}(C_i)$, $\Theta$ encapculates all parameters of MTG-Net, and $\Theta^*$ denotes a well-trained $\Theta$. By replacing the computation-intensive MTL procedure with a network inference, we can collect the estimations of all transferring gains much more efficiently. Moreover, as long as the estimation performance is reasonable, we are still able to identify near-optimal groupings.

---

**Algorithm 1:** Active Learning for MTG-Net

---

**Input:** $\mathcal{C}^{train} = \{C_{2^N-1}\}$, $\mathcal{C}^{test} = \mathcal{C} \setminus \mathcal{C}^{train}$, K, $\alpha$, $\eta$, Initialize $\Theta^*$, $\hat{\mathcal{Y}}^{test}$ via (2)

**for** $k$ **in** $[1, \cdots, K]$ **do**

    **for** $j$ **in** $[1, \cdots, N]$ **do**

        $\mathcal{C}^{T_j} = \{C_i,$ **for** $C_i \in \mathcal{C}^{test}$ **if** $T_j \in C_i\}$

        $\mathcal{G}^{T_j} = \{\hat{Y}_i(T_j),$ **for** $C_i \in \mathcal{C}^{T_j}\}$

        $\mathcal{P}^{T_j} = \{\exp^{\alpha \cdot |G_i|},$ **for** $G_i \in \mathcal{G}^{T_j}\}$    // Build the sampling distribution

        $C_{next} = \mathtt{Sample}_{\mathcal{P}^{T_i}}(\mathcal{C}^{T_i})$    // Select the next meta sample

        $\mathcal{C}^{train}.\mathtt{insert}(C_{next})$

        $\mathcal{C}^{test}.\mathtt{remove}(C_{next})$    // Update the meta-training set

        **if** $(k \cdot N - N + j) \% \eta == 0$ **then**

            Update $\Theta^*$, $\hat{\mathcal{Y}}^{test}$ via (2)    // Update MTG-Net and its predictions

        **end**

    **end**

**end**

**Output:** $\Theta^*$ and $\hat{\mathcal{Y}}^{test}$

---

## 3.4 Active Learning for MTG-Net

Given the meta-learning procedure (2) for MTG-Net, however, the construction of $\mathcal{C}^{train}$ is still a critical yet unsolved issue, which can have huge influences on the generalization capability of MTG-Net. To enable more efficient learning, we develop an active learning [35] strategy for MTG-Net. Here the key intuition is that paying more attention to the combinations with large gains can foster more efficient estimations of $\mathcal{Y}$ and also better facilitate the final grouping selection. Algorithm 1 includes the pseudo code of this strategy. First, we start with an MTG-Net trained on one meta sample that corresponds to the last task combination ($C_{2^N-1}$), which includes all $N$ tasks. Next, we conduct $K$ rounds of active selections. For each round, we repeat the same active learning process for all tasks. For each task ($T_j$), we first filter out a candidate set ($\mathcal{C}^{T_j}$) that incorporates all combinations including this task. Then for every combination ($C_i$) in $\mathcal{C}^{T_j}$, we collect the estimated transferring gain ($G_i = \hat{Y}_i(T_j)$) for this task, and we set the (unnormalized) sampling probability as $\exp^{\alpha \cdot |G_i|}$, where $\alpha$ is the hyper-parameter that measuring our preferences for remarkable gains (no matter positive or negative). Afterwards, we select a meta sample ($C_{next}$) following the distribution defined by $\mathcal{P}^{T_j}$, insert it into the meta-training set, and obtain updated $\Theta^*$, $\hat{\mathcal{Y}}^{test}$ by re-running the meta procedure for MTG-Net if meeting a given updating interval ($\eta$).

**Discussions on Efficiency**    Algorithm 1 includes $O(KN)$ times MTL procedures, which take up the major computational cost because the meta training in (2) takes negligible time compared with the corresponding MTL procedures to collect $\mathcal{Y}^{train}$. Besides, another issue that prevents from scaling to large $N$ lies in the exponential dependence of $|\mathcal{C}^{test}|$ on $N$. In practice, we can tackle this problem by randomly sampling a group of combinations with a large yet affordable size to substitute $\mathcal{C}$. Moreover, since the encoder part of a hard-sharing architecture usually takes up the most computation, the computational cost of $N$-task MTL procedures does not have orders of magnitude differences for different $N$. Thus in this work we employ the number of MTL procedures as an approximate yet intuitive measure of computational cost. While as an anonymous reviewer mentioned, if each task has its specific data, the real computational time of MTL procedures should be considered carefully.

## 4 Experiments

We conduct extensive experiments to validate the effectiveness and the efficiency of our method across diversified multi-task scenarios, including vision, energy, and healthcare. In this section, we only include basic setups, main experimental results, and a part of visualization analyses. Due to the space limit, we leave other details (datasets, task specifications, networks for MTL, hyper-parameters, etc.) and more results (more visualizations, case studies, etc.) to the appendix.

## 4.1 Experimental Setups

**MTL Datasets**   We conduct experiments on three MTL datasets. 1) Taskonomy [49] is a computer vision dataset including massive indoor scenes. Existing HOA-based methods [41, 14] leveraged 5 vision tasks of this dataset to evaluate MTG. We also follow the setups in [41] and denote the preprocessed dataset as `Taskonomy-5`. 2) ETTm1 [46] is an electric load dataset with 7 time series. We follow the multi-variate time-series forecasting problem in [46] and regard the forecasting for each series as a task. Accordingly, we denote this dataset as `ETTm1-7`. 3) MIMIC-III [17] is a healthcare database with rich electronic health records. Previous studies [30] constructed tens of clinical prediction tasks and found that simply performing MTL on these tasks did not always bring performance gains. We select 27 crucial clinical tasks to build the `MIMIC-III-27` dataset.

**Transferring Gains & Grouping Selection**   Section 3.1 introduces how to collect ground-truth transferring gains and how to perform the final grouping selection given transferring gains (or predictions). Here we make some detailed specifications for each MTL dataset. For 5 vision tasks on `Taskonomy-5`, we follow [41, 14] to use the loss value as the metric and thus calculate the transferring gain as the relative reduction of the loss. For 7 forecasting tasks on `ETTm1-7`, we instantiate the transferring gain as the relative reduction of the mean absolute error. While for 27 clinical prediction tasks on `MIMIC-III-27`, all of which adopt AUROC as the metric, we define the transferring gain as the relative improvement of AUROC. Note that all these transferring gains are collected on the validation set, and we conduct the MTL procedure for each case with different random seeds. Only after conducting the grouping selection based on transferring gains, we report the performance on the test set for selected groupings. Moreover, we collect all ground-truth transferring gains for 31 task combinations on `Taskonomy-5` and 127 task combinations on `ETTm1-7`. However, 27 tasks on `MIMIC-III-27` produce over one hundred million task combinations, so we randomly sample $3,000$ combinations to represent the whole space. All these MTL procedures cost thousands of GPU hours in total, and we will release the collected meta datasets for future research.

**Hyper-parameters of MTG-Net**   We use the same hyper-parameters for all MTL datasets. To be specific, we set the dimension of task embeddings as $D = 64$ and stack 2 self-attention encoding layers [45]. As for Algorithm 1, we set $\alpha$ as 25 to prioritize the selection of task combinations with large gains. Besides, we leverage a dynamic strategy to schedule $\eta$. At an early stage when $|\mathcal{C}^{train}| <= N + 1$, we set $\eta$ as 1 to frequently updating `MTG-Net` to pursue more effective selections. When $|\mathcal{C}^{train}| > N + 1$, we set $\eta$ as $N$ to reduce the number of updating for `MTG-Net` to further improve efficiency. $K$ is the hyper-parameter deciding the total number of meta-training samples, which is specified along with each figure. Moreover, we repeat the meta training and inference of `MTG-Net` with five random seeds and report the average results to eliminate the effects of randomness.

**Baselines**   We compare `MTG-Net` with four baselines. 1) `Oracle` denotes the grouping solution based on all ground-truth transferring gains on the validation set, which forms an upper bound of all approximation methods. 2) `Random` denotes randomly selecting a group of task combinations satisfying a given budget. Since this solution has high variance, we conduct one million random trials and report the average performance, as did in [41]. 3) `HOA` [41] conducted $\binom{N}{2}$ MTL procedures on pairwise combinations to collect ground-truth pairwise gains and then estimated all rest high-order gains via the average of corresponding pairwise gains. 4) `TAG` [14] attempted to improve the efficiency of HOA by obtaining an approximation to the pairwise gains via a customized MTL procedure on all tasks, which involved $N$ times extra forward and backward processes per each gradient updating to collect pairwise affinities on the fly.

## 4.2 Main Experiments

We present our main evaluation results for all three datasets in Figure 2. For each dataset, we include two configurations of $K$ for `MTG-Net`: $K = 1$ and $K = \frac{N-1}{2}$. As illustrated in Section 3.4, our active learning strategy requires $O(KN)$ times MTL procedures. Therefore, these two types of $K$ roughly correspond to the computational cost required by `TAG` and `HOA`, respectively.

In the left side of Figure 2, we show the final performance (on the test set) of selected groupings under different budgets. As the budget increases, meaning we have more combinations for a task to choose as the affiliated group, the final performance should also increase for a well-behaved

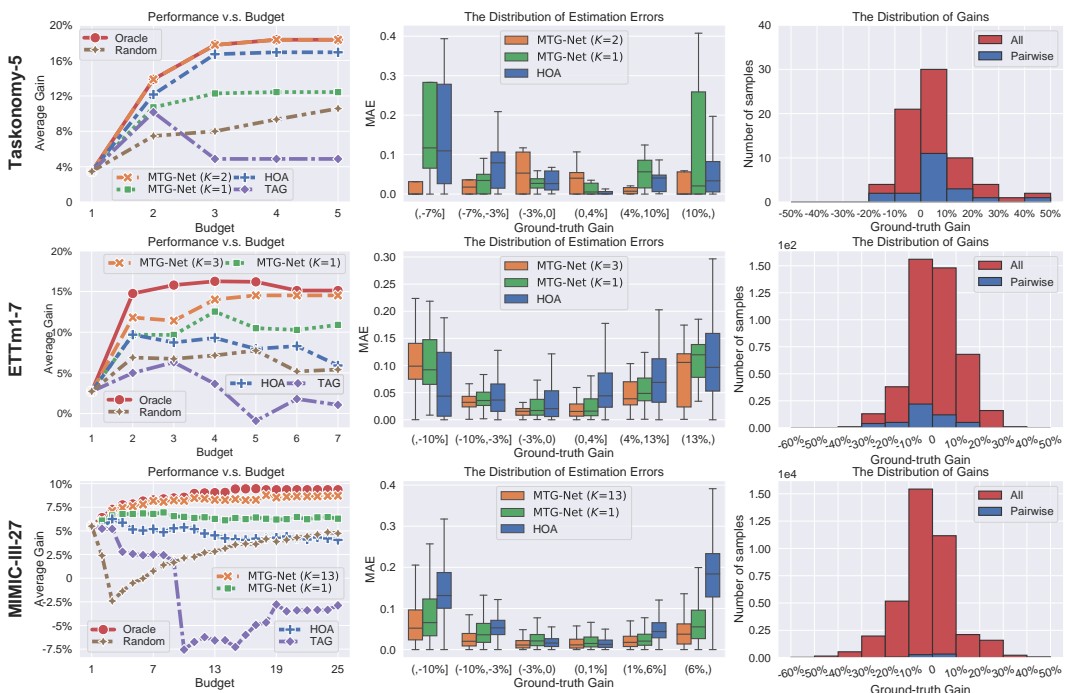

Figure 2: We show the final performance of selected groupings under different budgets (left column), the distribution of gain estimation errors (middle column), and the distribution of ground-truth gains (right column) on `Taskonomy-5`, `ETTm1-7`, and `MIMIC-III-27`.

MTG solution. However, we observe some fluctuation and even degradation effects. There are two reasons: 1) inaccurate estimations of transferring gains can largely misguide the grouping selection, and the grouping errors become much more evident as the budget increases; 2) we use the transferring gains collected on the validation set to guide the grouping selection, but these gains are not completely consistent with the actual performance gains on the test set. The latter reason also apply to `Oracle`, but its performance increases with the budget at most time. Thus, we can conclude that the degradation effect influenced by the latter reason is limited, and it is reliable to attribute the final grouping performance to the quality of the estimations for transferring gains.

Furthermore, we can find that `HOA` and `TAG` produce poor grouping decisions on `ETTm1-7` and `MIMIC-III-27`, while `HOA` can obtain pretty well grouping performance on `Taskonomy-5`, and `TAG` can produce reasonable performance when the budget is small. The key reason is that the number of task combinations increases exponentially as the number of tasks increases, as a result, `HOA` and `TAG` fail to provide accurate estimations for massive high-order task combinations solely based on the information of pairwise combinations. As for `Taskonomy-5`, there are large transferring gains covered by pairwise combinations, so `HOA` can obtain reasonable grouping performance. However, since `TAG` introduces another approximation step to predict pairwise affinities, its grouping performance suffers more degradation than `HOA` as the budget increases. In contrast, we can see that `MTG-Net` behaves much better on all cases. Specifically, when we set $K = \frac{N-1}{2}$, which means we only have the ground-truth gains on $\frac{N(N-1)}{2} + 1$ task combinations, `MTG-Net` is able to produce high-quality groupings that are very close to the ones derived by `Oracle`. In addition to the grouping performance, we also show the estimation errors of transferring gains (calculated on the validation set) on all task combinations in the middle column of Figure 2 and plot the distribution of ground-truth transferring gains in the right side. These auxiliary information aligns with the analyses mentioned above and further reveal the drawbacks of the HOA assumption that maps pairwise gains to high-order ones.

Given the superiority of `MTG-Net` over existing solutions, we further conduct experiments on the much challenging `MIMIC-III-27` benchmark to verify two critical factors that drive the success of `MTG-Net`. One is the number of meta-training samples. Figure 3a includes the performance comparisons of multiple `MTG-Net` counterparts with different $K$. Here we conduct multiple runs for each counterpart and plot the average performance as well as the standard deviation (shadow

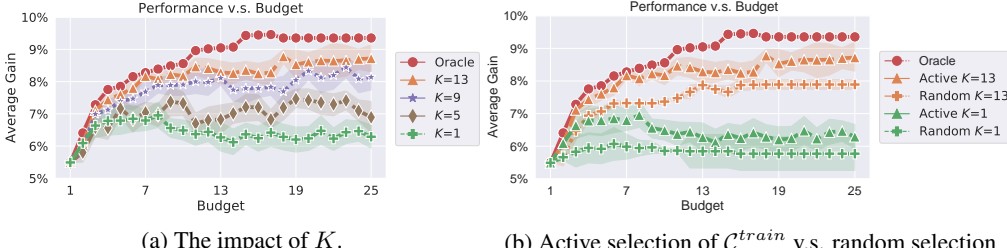

(a) The impact of $K$.  (b) Active selection of $\mathcal{C}^{train}$ v.s. random selection

Figure 3: Different configurations of `MTG-Net` on `MIMIC-III-27`.

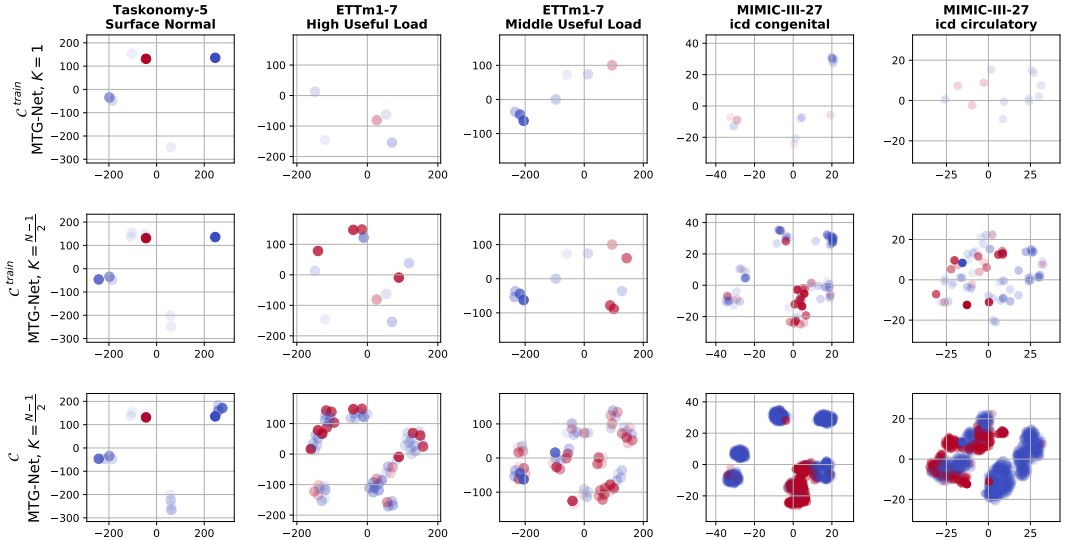

Figure 4: We select five tasks from different datasets and visualize the latent structures of transferring relationships discovered by `MTG-Net`. For each task, we aggregate its encoded embeddings (from $\boldsymbol{H}_i$) for all relevant task combinations and project them into a two-dimensional space via t-SNE [44]. The upper two rows only include task combinations on $\mathcal{C}^{train}$ for two $K$s. The bottom row includes all combinations covering this task. We color each combination by its ground-truth gain for this task (red: positive; blue: negative; the deeper the color depth, the larger the gain (absolute value)).

area). We can clearly see that adding meta-training samples brings prominent improvements in the grouping performance for large budgets. While for small budgets, even $K = 1$ can produce reasonably good performance. This observation can help us to balance efficiency and effectiveness flexibly according to specific practical scenarios. Besides, the other crucial factor is the active learning strategy illustrated in Algorithm 1. Figure 3b includes the ablation study on the importance of active learning. We can observe that no matter $K$ is large (13) or small (1), the active selection of meta samples produces remarkable improvements over the random selection.

### 4.3 Visualization Analyses

Figure 4 visualizes some latent structures of transferring relationships discovered by `MTG-Net`. By inspecting the upper two rows, we can clearly see that `MTG-Net` discovers a more comprehensive latent structure as the active learning strategy progressively increases $|\mathcal{C}^{train}|$ from $N + 1$ to $\frac{N(N-1)}{2} + 1$. From the bottom two rows, we can observe that the latent structure discovered on $\mathcal{C}^{train}$ is roughly consistent with the overall structure on $\mathcal{C}$. This observation intuitively reveals that the transferring relationships indeed lie in some low-dimensional manifolds, and our active learning strategy is effective in selecting those crucial combinations, which can function as the landmarks of the latent structure. Moreover, by comparing different latent structures across different tasks, we can observe that no matter how the MTL scenario and the number of task combinations vary, `MTG-Net` can still provide a reasonable approximation to the underlying latent structure.

# 5 Conclusion and Future Work

This paper introduces an efficient and effective approach for MTG, which is based on a new meta-learning formulation on task combinations. The key takeaway is that no matter how large the space of task combinations is, the relationships between task combinations and performance gains lie in some low-dimensional manifolds. Our experiments across diversified MTL scenarios demonstrate the practicability of this hypothesis. This is why we can make predictions for massive unseen task combinations with only ground-truth gains on a few actively selected combinations. The direct impact of this work is to benefit a wide range of real-world applications [6, 19, 36, 31], in which people have multiple learning tasks but do not know how to organize them into different groups effectively.

Moreover, we note that there are some valuable future research directions given the contribution of this work, demonstrating that the transferring relationships of a predefined MTL procedure across different task combinations could be meta learned. First, when the data or the model architectures changed, the transferring effects among tasks could also distinctly change. Both HOA [41] and us have observed these phenomena, which implies that multi-task transferring relationships may be a function of the data, the model, and some other factors (such as optimization algorithms). Thus how to enable effective meta learning on task combinations across different configurations of MTL procedures is a valuable research question to be answered. Besides, this work only considers the transferring relationships among a fixed set of tasks, while in practice we would always encounter new tasks. Accordingly, how to enable incremental meta learning or endow `MTG-Net` with the zero-shot capability is worthy of great research attention.

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
