# OpenReview forum: "Efficient and Effective Multi-task Grouping via Meta Learning on Task Combinations"
_NeurIPS.cc/2022/Conference — NeurIPS 2022 Accept_

### Official Review · Reviewer_FUf3 · 2022-07-09

**Rating:** 5
**Confidence:** 5
**Soundness:** 2 fair
**Presentation:** 3 good
**Contribution:** 3 good

**Summary:**

This paper researches on the task grouping problem in Multi-task Learning (MTL). Task grouping matters MTL’s performance and has gained much attention recently. This paper proposes a novel method to address the task grouping problem, which formulate  a meta-learning problem for task grouping. The proposed method has achieved superior performance.

**Questions:**

What’s the performance of the active learning strategy in the early epochs?

**Ethics Review Area:**

["I don’t know"]

**Limitations:**

The technical contribution of this paper is limited for it just utilizes the well-known meta learning frame-work.

**Strengths And Weaknesses:**

Pros

1.The proposed method is experimentally verified to be effective on various applications.
2.The paper has conducted extensive experiments .
3.The paper is easy to follow.

Cons

1.The technical contribution of this paper is limited for it just utilizes the well-known meta learning frame-work.

---

> ### Author Response · Authors · 2022-08-02
> **Response to Reviewer FUf3**
>
> Thanks for your efforts in reviewing this paper.
>
> Regarding your concern of “the technical contribution is limited”, we would like to emphasize a few unique technical contributions of this paper, as elaborated below, beyond a simple application of existing meta-learning framework:
>
> - In the first place, this paper reveals a critical finding: the relationships between task combinations and transferring gains are essentially some low-dimensional manifolds. This finding stands as the foundation for building the meta learning framework for multi-task grouping with high generalization capability.
> - Furthermore, this paper recognizes a new yet critical meta-learning problem on task combinations, i.e., learning to predict the transferring gains given a task combination.
> - To address this problem, this paper introduces a novel meta model, MTG-Net, that is capable of being generalized to unseen task combinations simply by taking task tokens as the input.
> - To further ensure the efficiency of this framework, this paper introduces an active-learning algorithm that is used to facilitate more efficient explorations of the exponentially growing space of task combinations.
>
> Further indication about the performance of the active learning algorithm in the early epochs can be found by inspecting Figure 3 (a, b), in which we plot the results of MTG-Net under different $K$s. Specifically, $K$=1 denotes a very early epoch, and we can observe that MTG-Net can still provide some useful estimations of transferring gains for the grouping selection algorithm to obtain reasonable performance. Besides, you can find the convergence of MTG-Net under different $K$s in Section D.3 of the appendix. These results indicate fast convergence of MTG-Net in early epochs.

---

### Official Review · Reviewer_q6pk · 2022-07-12

**Rating:** 7
**Confidence:** 3
**Soundness:** 3 good
**Presentation:** 4 excellent
**Contribution:** 3 good

**Summary:**

The paper proposes a method for multi-task grouping, which is to find the best sets of multi-task networks that covers the given target tasks. Specifically, the paper trains a model that predicts performance gains as a function of task combinations. Based on this prediction, best task combinations that maximizes the total performance gain can be obtained for given N tasks and budget B through the grouping selection algorithm. To save the training cost, the paper proposes an active learning algorithm that selects task combinations to use for training based on the predicted performance gains. Experiments show that the proposed method outperforms existing MTG methods and predicts total performance gains reasonably well on three MTL datasets (Taskonomy-5, ETTm1-7, MIMIC-III-27).

**Questions:**

- It would be interesting to see which task combinations are selected at each step of active learning. How similar they are for each task and do their sizes change over iterations? Also, it would be interesting to see the average train/test error of gain predictions over the iterative training combination sets selection procedure, how it improves and what is the final error.
- How sensitive the method is to the different design choices of MTG-Net (initialization, number of layers, feature dimension, etc)? Since it is trained using only a small sets of (task combination, accuracy gains) pairs, the MTG-Net design can affect a lot on the prediction accuracy of the performance gains. For example, at the first round of MTG-Net training (k=1 in Algorithm1), the MTG-Net used to sample meta-training set is trained using only one (task combination, accuracy gain)-pair. I am curious how much they affect the results.
- How does the method work for different hyperparameter choices (D, alpha, eta)? For example, alpha is currently set to 25, which I guess will greedily choose the task combinations with best gain for each task. This makes sense to me because these task combinations with extreme gains likely serve as anchors and help interpolating gains for other combinations. At the same time, selecting diverse combinations could be more sample-efficient when training MTG-Net. How important the current hyperparameter choices are and what happens they are set to other values?
- It would be interesting to see if task relations are model-agnostic. Will the gain estimation obtained from smaller MTL backbone generalizes to larger MTL backbones? This may reduce the training cost.
- Which task prediction is selected at test time if that task is included in several task combinations and there are multiple predictions?


**Limitations:**

The authors discussed the limitations in the paper. I do not see potential negative societal impact.

**Strengths And Weaknesses:**

I think this paper proposes an interesting approach for the multi-task grouping problem. Compared to [41] which also takes into account the compute budget constraint, this paper does not consider the compute budget in a strict manner, however, the computational complexity is reduced from O(2^n) to O(n^2).

1. Strengths
- The key finding of the paper is that the relationships between task combinations and performance gains lie in some low-dimensional space and can be approximated reasonably well by the proposed self-attention encoder architecture. I think this is an original finding and also provides good insights.
- Unlike existing works which approximated relations of task combinations using pairwise task relations, this paper proposes to predict the task gains as functions of higher order task combinations directly using the self-attention encoder and task representations. I think this design is novel and reasonable.
- The paper extensive experiments on three datasets, ranging from 5 to 27 tasks. Ground truth performance gains are collected and will be released, which I believe will be useful for future MTL research.

2. Weaknesses
- I think one weakness is the high computational cost. The method requires multi-task network training for O(NK) task combinations. Though it is much smaller than the number of all possible combinations, however, this could be still expensive especially when the number of tasks is large or datasets are large.
- Analysis of the effect of some design choices (hyperparameters, MTL model architecture, MTG model design) is missing.
- As also mentioned in the paper, the prediction accuracy is less reliable with smaller number of meta-training data selection. Since MTG-Net is trained using only a small sets of (task combination, accuracy gains) pairs, the MTG-Net design (initialization, architectures) can affect a lot on the prediction accuracy of the performance gains. For example, at the first round of MTG-Net training (k=1 in Algorithm1), the MTG-Net used to sample meta-training set is trained using only one (task combination, accuracy gain)-pair.
- It is good that the paper takes into account the budget B, which is defined as the allowed number of task groups. However, the actual resource usage  (e.g. inference time, memory), which are of more interest in practice, may not be proportional to this budget B because the multi-task network for each group may have different resource usages.
- Experiments look relatively weak in terms of dataset size and task diversity. It would be nice to have experiments for datasets with larger number of data points and tasks.

---

> ### Author Response · Authors · 2022-08-02
> **Response to Reviewer q6pk (Part 5/5)**
>
> In addition to answering your specific questions, we want to provide more discussions on other weaknesses you have mentioned.
>
> **Response to Weakness 1**
>
> The huge computational cost is indeed the biggest challenge in the multi-task grouping problem. Essentially, such huge computational cost can be divided into two folds: one is the exponential dependence on the number of tasks, and the other is the cost per each MTL procedure. One of the major contributions of this paper lies in addressing the former aspect. The key contribution of this paper is to reduce the complexity of conducting MTL procedures from $O(2^N)$ to $O(KN)$ ($K$ is an adjustable hyper-parameters) while still maintaining reasonably well performance. While in previous studies [1, 2], they relied on some predefined heuristics, which resulted in biased estimations for high-order transferring gains. For larger datasets and much more tasks, as long as the computational resources allow us to perform $O(KN)$ rounds of MTL procedures (K < 1 may also work), our framework can provide much better grouping options than random sampling or other heuristic-based approaches.
>
> **Response to Weakness 2 & 3**
>
> We really appreciate your suggestion of adding more hyper-parameter analyses. All the supplemented experimental results have been provided in the responses to Questions 2 and 3, and we will also add them to the revised paper. We hope these results can help to address your concerns on “missing hyper-parameter analyses” and “instability of MTG-Net learning due to very limited meta samples”.
>
> **Response to Weakness 4**
>
> We totally agree with you that some actual resource usages, such as inference time and memory, are of more interest in practice. In fact, it is relatively straightforward to combine our framework with these practical considerations. For example, we can conduct the MTL procedures of multiple models with different inference time and memory consumptions and then feed a tuple of (a task combination, inference time, memory used, the associated transferring gains) to an adjusted grouping selection algorithm for decision making. While this paper mainly focuses on the most fundamental setup (e.g., exemplifying the budget as the number of groupings) to highlight our critical findings and key ideas, we will continue the research under practical considerations as the future work.
>
> **Response to Weakness 5**
>
> We demonstrate the effectiveness of our framework over three datasets with diversified tasks, including not only indoor-scene recognition tasks used by [1, 2] but also energy time-series regression tasks as well as clinical classification tasks. Moreover, the setups of the number of tasks also cover the small (5), moderate (7), and large (27) scales. Specifically, 27 tasks can produce over one hundred million task combinations, and none of the existing studies has seriously studied the multi-task transferring relationships at this scale. Moreover, due to the curse of combinatorial explosion, we have taken thousands of GPU hours in total to collect the transferring gains for all task combinations (except for MIMIC-III-27, we sample 3000 task combinations to approximate the huge space). Such extensive experiments on various tasks have provided very solid verification on the effectiveness of the critical ideas and new MTG framework proposed by this paper. We have a strong belief that all the thorough findings together with the novel framework can benefit much to the whole MTL community.
>
> [1] Standley, Trevor Scott et al. “Which Tasks Should Be Learned Together in Multi-task Learning?” ICML (2020).
>
> [2] Fifty, Chris, et al. "Efficiently identifying task groupings for multi-task learning." NeurIPS (2021).

---

> > ### Comment · Reviewer_q6pk · 2022-08-08
> > **Thank you for the detailed answers.**
> >
> > Thank you for the detailed answers. I would like to keep my score.

---

> ### Author Response · Authors · 2022-08-02
> **Response to Reviewer q6pk (Part 4/5)**
>
> **Response to Question 4**
>
> We find that these transferring relationships among tasks are not model agnostic, and this finding is consistent with the phenomenon observed by HOA [1].
>
> One of the experiments in [1] was to compare two kinds of MTL procedures on the same dataset, in which the only difference was the model size (a large Xception network v.s. a small one). According to their results, the transferring gains collected from these two types of models were rather different.  The correlation coefficient is merely around 0.23 (calculated by us based on their released results). We also have tried other architectures on ETTm1 and MIMIC-III-27 and obtained similar observations. Moreover, these task relations are not data agnostic (Experiments in [1] also demonstrated this claim), which means that when the data distribution or the data scale changes, we may obtain different transferring gains.
>
> Therefore, HOA, TAG, and our work are all studying a “static” multi-task grouping problem, in which the setup of the MTL procedure is fixed (data, model, optimization, etc.). In such a “static” setup, our method contributes to reducing the complexity of performing MTL procedures from $O(2^N)$ to $O(KN)$.
>
> In the meanwhile, we think this question is very interesting and inspiring because it points out a new direction worthy of more research attention, which is to study the transferring relationships across MTL model architectures. (Please kindly note that this question is also related to the 4-th question raised by Reviewer ZASQ.)
>
> **Response to Question 5**
>
> This question is more related to the grouping selection algorithm. If a task is involved in multiple groupings, the grouping selection algorithm will assign its inference group to the one with the largest transferring gains (or gain predictions).
>
> You may refer to the HOA paper [1] and its appendix (http://proceedings.mlr.press/v119/standley20a/standley20a-supp.pdf) for more details about the grouping selection algorithm.
>
> [1] Standley, Trevor Scott et al. “Which Tasks Should Be Learned Together in Multi-task Learning?” ICML (2020).

---

> ### Author Response · Authors · 2022-08-02
> **Response to Reviewer q6pk (Part 3/5)**
>
> **Response to Question 3**
>
> Like the setup of MTG-Net’s hyper-parameters discussed in the response to Question 2, we also employ a modest choice of $\alpha$ and $\eta$ in the active learning procedure, and our framework can still work well with other options. In the following, we provide concrete results and detailed explanations to give you more insights.
>
> First, we strongly agree with your intuition on using a relatively large $\alpha$, which stimulates MTG-Net to prefer task combinations with larger gains and then leverage them as anchors to interpolate gains for other combinations. Below we include the max-budget grouping performance of MTG-Net with different $\alpha$s to reveal the impact of $\alpha$.
>
> |Dataset | Meta Model| $\alpha$=0.01 | $\alpha$=0.1 | $\alpha$=1 | $\alpha$=5 | $\alpha$=10 | $\alpha$=25 | $\alpha$=100 |
> | --- | ---| ---| ---| ---| ---| --- | ---| --- |
> |Taskonomy-5| MTG-Net (K=1) |+11.35% |+11.50% |+11.95% |+12.35% |+12.43% |+12.43%|+12.43%|
> |Taskonomy-5| MTG-Net (K=2) |+17. 75%|+17.90% |+18.14% |+18.20% |+18.34% |+18.34%|+18.34%|
> |ETTm1-7| MTG-Net (K=1) |+10.11% |+10.15% |+10.52% |+10.57% |+10.61% |+10.61%|+10.61%|
> |ETTm1-7| MTG-Net (K=3) |+14.30% |+14.37% |+14.46%|+14.50%|+14.52% |+14.52%|+14.52%|
> |MIMIC-III-27| MTG-Net (K=1) |+5.86%|+6.07%|+6.15%|+6.23%|6.26%|+6.28%|+6.28%|
> |MIMIC-III-27| MTG-Net (K=13) | +7.91%|+8.13%|+8.61%|+8.65%|+8.70%|+8.73%|+8.74%|
>
> We can observe that MTG-Net can still obtain reasonable performance with other setups of $\alpha$, and by significantly preferring task combinations with larger gains, MTG-Net is able to obtain slightly more improvements in terms of the final grouping performance. The rationale behind this observation is that the estimation errors on larger transferring gains can cause more direct error propagations to the grouping selection algorithm, thus paying more attention to these estimation errors help most in the final grouping performance (this point aligns with a similar question raised by Reviewer QRvu).
>
> Moreover, the hyper-parameter $\eta$ controls the frequency of updating MTG-Net with actively selected meta samples, which allows a trade-off between efficiency and effectiveness for MTG-Net training. In this work, we adopt a dynamic strategy: 1) in the earliest round (k = 1), we set $\eta$ as 1 to encourage more frequent MTG-Net updating per each active selection, 2) and in latter epochs (k > 1), we set $\eta$ as N to further improve the efficiency by updating MTG-Net per N active selections. In the following table, we compare this dynamic strategy with two fixed strategies ($\eta$=1 or $\eta$=N) to reveal the impact of $\eta$.
>
> | Meta Model | $\eta$ | # MTG-Net Updating Steps| Taskonomy-5 | ETTm1-7 | MIMIC-III-27 |
> | --- | ---| ---| ---| ---| ---|
> | MTG-Net (K=1) | $\eta$ = 1 | K*N $\rightarrow$ N |+12.47% | +10.65%| +6.29%|
> | MTG-Net (K=1) | $\eta$ = N | K $\rightarrow$ 1 |+11.27% | +9.97%| +5.97%|
> | MTG-Net (K=1) | $\eta$ = 1 if k <=1 else N | N+K-1 $\rightarrow$ N |+12.43% | +10.61%|+6.28% |
> | MTG-Net (K=(N-1)/2) | $\eta$ = 1 | K*N $\rightarrow$ N*(N-1)/2 |+18.34% |+14.55% |+8.75% |
> | MTG-Net (K=(N-1)/2) | $\eta$ = N | K $\rightarrow$ (N-1)/2 |+17.75% | +14.26%| +8.69%|
> | MTG-Net (K=(N-1)/2) | $\eta$ = 1 if k <=1 else N | N+K-1 $\rightarrow$ (N-1)*3/2  |+18.34% |+14.52% |+8.73% |
>
> The above results demonstrate that our dynamic strategy for $\eta$ can help MTG-Net with different $K$s achieve competitive performance as setting $\eta$ as 1 while ensuring that the number of MTG-Net updating steps is not an order of magnitude larger than that of setting $\eta$ as $N$.

---

> ### Author Response · Authors · 2022-08-02
> **Response to Reviewer q6pk (Part 2/5)**
>
> **Response to Question 2**
>
> We very appreciate your suggestion of adding more hyper-parameter analyses to make this work more promising. In the following, we provide detailed explanations and supplement more hyper-parameter testing results for your reference. Overall speaking, our framework is quite robust against varying hyper-parameters of MTG-Net. From another point of view, such robustness can imply that a fixed set of hyper-parameters, as presented in our paper, can work consistently well across three different datasets with diversified tasks.
>
> Regarding the sensitivity to the initialization of MTG-Net, Figures 2, 3 in the paper already revealed some insights since we re-run the MTG-Net training with five different random seeds and report the average results there. Please note that in Figure 3 (a, b), we also include the standard deviation (shadow area). These results can validate that MTG-Net is robust to different initializations.
>
> Moreover, MTG-Net is also robust to different configurations of the number of encoding layers and the hidden dimension (D), which in this paper are set as 2 (#layers) and 64 (D), respectively. Below we show the max-budget grouping performance of multiple MTG-Net models with different setups of these two hyper-parameters on MIMIC-III-27.
>
> | Dataset | Meta Model | #layers = 1|#layers = 2|#layers = 3|#layers = 4|#layers = 5|
> | --- | ---| ---| ---| ---| ---| --- |
> |Taskonomy-5| MTG-Net (K=1) |+12.17% |+12.43% |+12.43% |+12.35% |+12.35% |
> |Taskonomy-5| MTG-Net (K=2) |+17.90% |+18.34% |+18.34% |+17.90% |+18.34% |
> |ETTm1-7| MTG-Net (K=1) |+10.52% |+10.61% |+10.61% |+10.57% |+10.52% |
> |ETTm1-7| MTG-Net (K=3) |+14.44% |+14.52% |+14.50% |+14.44% |+14.50% |
> |MIMIC-III-27| MTG-Net (K=1) | +6.25% | +6.28% | +6.29% | +6.29% | +6.31% |
> |MIMIC-III-27| MTG-Net (K=13) | +8.63% | +8.73% | +8.74% | +8.76% | +8.78% |
>
> | Dataset | Meta Model | D=8 | D=16 | D=32 | D=64 | D=128 |
> | --- | ---| ---| ---| ---| ---| --- |
> |Taskonomy-5| MTG-Net (K=1) |+12.35% |+12.43% |+12.43% |+12.43% |+12.35% |
> |Taskonomy-5| MTG-Net (K=2) |+17.90% |+18.34% |+18.34% |+18.34% |+18.34% |
> |ETTm1-7| MTG-Net (K=1) |+10.52% |+10.59% |+10.61% |+10.61% |+10.61% |
> |ETTm1-7| MTG-Net (K=3) |+14.42% |+14.50% |+14.50% |+14.52% |+14.52% |
> |MIMIC-III-27| MTG-Net (K=1) | +6.07% | +6.10% | +6.17% | +6.28% | +6.32%|
> |MIMIC-III-27| MTG-Net (K=13) | +8.12% | +8.45% | +8.62% | +8.73% | +8.79%|
>
> We have a few interesting observations from the two tables above.
>
> First, we find that different setups of the number of layers and the hidden dimension can all give rise to significant and consistent improvements in the final grouping performance. This observation can imply that our framework is robust to the hyper-parameters of MTG-Net.
>
> Second, we can observe that some different setups even lead to the exact same grouping performance, especially on Taskonomy-5 and ETTm1-7 (the number of task combinations is relatively small). The underlying reason is that although these different MTG-Net models produce different gain predictions, these predictions can be consistent in selecting task combinations for a specific task, especially given limited combinations, and thus lead to the same grouping result.
>
> Besides, we observe certain correlations between the model capacity and the grouping performance. For example, as # layers or D increases, which indicates the increasing of model capacity, we usually observe associated improvements in grouping performance. This observation indicates that there exist some highly non-linear transferring relationships in the space of task combinations. Since the increasing of model capacity also enlarges the risk of overfitting, we also observe some performance dropping for some large values of #layers and D.
>
> Moreover, we can see that some setups of hyper-parameters (e.g., D=128) even slightly outperform the one presented in this paper. This observation implies that we do not overly fine-tune the hyper-parameters of MTG-Net, and the setup presented in this paper is a modest choice.

---

> ### Author Response · Authors · 2022-08-02
> **Response to Reviewer q6pk (Part 1/5)**
>
> Thanks so much for all your thorough reviews, insightful comments, and many constructive suggestions, based on which we will further enrich the technical and experiment content of this paper. Besides, we hope the following responses can help answer your questions and address your concerns.
>
> **Response to Question 1**
>
> In accordance with your suggestion, we will append the detailed task combinations selected by MTG-Net at each step of active learning into the revised appendix to ease understanding. Here we provide a few snapshots of the active learning procedure as a quick response. (Please refer to the appendix to align task names with task indicators.)
>
> |Dataset| Round k | Task j | Selected Combinations | Ground-truth Gain | Predicted Gain Before Selection | Predicted Gain After Selection|
> | --- | ---| ---| ---| ---| ---| --- |
> |Taskonomy-5|1|$s$|[$s$,$k$]|6.3%|1.6%|3.2%|
> |Taskonomy-5|2|$s$|[$s$,$d$,$n$,$e$]|7.2%|15.9%|9.1%|
> |Taskonomy-5|1|$n$|[$s$,$d$,$n$]|-16.7%|-3.4%|-5.1%|
> |Taskonomy-5|2|$n$|[$n$,$e$]|1.6%|-4.5%|0.2%|
> |ETTm1-7|1| HF |[HF,MF,LL]|-5.5%|-2.3%|-6.5%|
> |ETTm1-7|3| HF |[HF,ML,LF]|19.2%|5.4%|7.3%|
> |ETTm1-7|1| LL |[LL,OT]|-3.5%|-5.5%|-2.8%|
> |ETTm1-7|3| LL |[HF,HL,MF,LF,LL]|-2.0%|4.1%|-3.5%|
> |MIMIC-III-27|1|$t_1$|[$t_{1}$,$t_{2}$,$t_{3}$,$t_{5}$,$t_{8}$,$t_{9}$,$t_{10}$,$t_{16}$,$t_{17}$,$t_{19}$,$t_{21}$,$t_{23}$,$t_{24}$,$t_{25}$]|-8.0%|-2.0%|-7.3%|
> |MIMIC-III-27|7|$t_1$|[$t_{1}$,$t_{3}$,$t_{6}$,$t_{7}$,$t_{9}$,$t_{10}$,$t_{11}$,$t_{14}$,$t_{16}$,$t_{26}$,$t_{27}$]|4.4%|6.8%|5.7%|
> |MIMIC-III-27|13|$t_1$|[[$t_{1}$,$t_{12}$,$t_{13}$,$t_{14}$,$t_{15}$,$t_{17}$,$t_{18}$,$t_{19}$,$t_{24}$,$t_{26}$]|-7.7%|-9.3%|-8.4%|
> |MIMIC-III-27|1|$t_{24}$|[$t_{4}$,$t_{8}$,$t_{9}$,$t_{12}$,$t_{13}$,$t_{14}$,$t_{17}$,$t_{20}$,$t_{24}$]|-3.2%|-36.1%|-5.1%|
> |MIMIC-III-27|7|$t_{24}$|[$t_{8}$,$t_{10}$,$t_{11}$,$t_{13}$,$t_{15}$,$t_{19}$,$t_{21}$,$t_{22}$,$t_{23}$,$t_{24}$,$t_{25}$]|20.1%|20.7%|20.5%|
> |MIMIC-III-27|13|$t_{24}$|[ $t_{1}$,$t_{2}$,$t_{3}$,$t_{4}$,$t_{7}$,$t_{8}$,$t_{9}$,$t_{11}$,$t_{12}$,$t_{16}$,$t_{20}$,$t_{22}$,$t_{23}$,$t_{24}$ ]|21.1%|23.5%|22.3%|
>
> From the table above, we can observe diversified combinations being selected for different tasks at different rounds. More importantly, these results intuitively reveal that MTG-Net can calibrate its predictions after updating with the ground-truth gains.
>
> Besides, we do not find a clear pattern of the variation in terms of the size of selected task combinations as the active learning process iterates. The sizes of selected task combinations seem to be consistent across different Ks. The following table includes mean$\pm$std of the size of selected task combinations for each round (K) on all datasets.
>
> |Dataset| K=1 | K=2 | K=3| K=5| K=7| K=9| K=11| K=13|
> | --- | ---| ---| ---| ---| ---|--- |--- |--- |
> |Taskonomy-5|2.6$\pm$0.5|2.8$\pm$0.8 | - | - | - | - | - | - |
> |ETTm1-7|3.3$\pm$1.0|3.1$\pm$0.6|3.0$\pm$0.8| - | - | - | - | - |
> |MIMIC-III-27| 12.1$\pm$1.0 | 11.8$\pm$0.5 | 12.1$\pm$0.6 | 12.4$\pm$0.9 | 12.7$\pm$0.6 | 12.9$\pm$0.7 | 12.4$\pm$0.9 | 12.1$\pm$0.5 |
>
> Moreover, you can check the convergence of test errors on gain predictions in Section D.3 of the appendix. It is intuitive that MTG-Net converges very fast in early iterations and can still obtain relatively slight yet robust improvements as $K$ gets close to $(N-1)/2$. Besides, in the table below, we show the final estimation errors (measured in Mean Absolute Error) of MTG-Net on all datasets for your reference.
>
> | Meta Dataset | Taskonomy-5 | ETTm1-7 | MIMIC-III-27 |
> | --- | ---| ---| ---|
> | Meta Train | 0.068 | 0.044 | 0.045 |
> | Meta Test | 0.083 | 0.073 | 0.057 |
>
> We can observe that there is no overfitting issue, and MTG-Net can obtain reasonable generalization on massive unseen task combinations.

---

### Official Review · Reviewer_QRvu · 2022-07-15

**Rating:** 7
**Confidence:** 3
**Soundness:** 3 good
**Presentation:** 3 good
**Contribution:** 3 good

**Summary:**

This paper introduces a simple yet effective meta-learning framework for multi-task grouping. The main goal is to avoid computing the performance gains of multi-task learning (MTL) over single-task learning (STL) for all possible task combinations, which is intractable for a large number of tasks. During meta-training, a mapping function between task combinations and performance gains is learned, where the meta-training set is selected using an active learning strategy to keep it small in size while still being effective. During meta-testing, the performance gains of the rest of the task combinations are predicted using the learned mapping function. Finally, the task groupings are selected based on the meta-training (ground-truth) and meta-testing (predicted) performance gains. Experiments on 3 datasets demonstrate the effectiveness of the proposed method.

**Questions:**

I think this is a rather decent submission with a novel idea and convincing experimental results. Apart from the whopping computational cost of conducting the MTL procedures and perhaps the difficulty to scale to deeper models, I only have one minor suggestion for the authors to improve the paper.

- The proposed active learning strategy assigns sample weights to prioritize task combinations with larger (absolute) performance gains. According to my understanding, here the performance gains are computed using the learned MTG-Net (instead of ground-truth). Is there a rationale behind why doing so (i.e., prioritizing task combinations with larger predicted gains) is more helpful for learning MTG-Net?

I also identify the following typos.
- Line 196, “work tokens” should be “word tokens”.
- Line 198, the times sign $\times$ in the shape of $\mathbf{X}_i$ is written as an $x$.
- Line 205, the shape of $H_i$ should be $|C_i| \times D$.
- In Figure 1, the STL procedure in the left subfigure, you may change the title to "for $t^i_j \in C_i$" to keep the notation consistent with your text in line 132.
- In Figure 2, performance vs budget for Taskonomy-5, the oracle line is missing.


**Limitations:**

The authors have pointed out the limitations in lines 353-356.

**Strengths And Weaknesses:**

Strengths
- The proposed framework to learn the mapping between task combinations and performance gains is novel.
- The contribution is quite significant as based on their experiments, the proposed framework significantly improves over the pairwise non-parametric approaches under similar computational costs. The visualization also helps to reveal why it works.
- The paper is well-written and very easy to follow.
- Most of the presentations are clear.

Weaknesses
- Certain parts of the paper lack clarity (see suggestions below).
- The idea of selecting task groupings based on performance gains is very costly in nature as it involves a lot of MTL training. Given that the experiments cost “thousands of GPU hours in total” (line 268), the experimental results might be difficult to reproduce.

---

> ### Author Response · Authors · 2022-08-02
> **Response to Reviewer QRvu**
>
> Thanks very much for your insightful comments, careful inspection, and all the endorsement of this submission. We will fix all the typos in the revised manuscript. Besides, we hope the following explanations may address the other two concerns.
>
> **Response to Question 1 & Weakness 1**
>
> The rationale behind preferring meta samples with large transferring gains is that the estimation errors on larger transferring gains would cause more direct error propagations to the grouping selection algorithm. For example, if the best combination for a specific task identified by MTG-Net produces a moderate or even negative transferring gain, the grouping selection algorithm will pick this combination if the budget allows but make a serious mistake in fact. Thus, we should be careful about making errors on task combinations with large values of predicted transferring gains. Thanks again for posting this suggestion, we will further polish the sentences in the revised version to make this point much clearer.
>
> **Response to Weakness 2**
>
> To ensure the reproducibility of this work, we will not only release our data and codes but also take several extra steps.
>
> Specifically, in addition to opening the results of MTL procedures and the code of MTG-Net to help reproducing the results in this paper, we will open the specific code to conduct MTL procedures, which can help other researchers double confirm the transferring gains of our MTL procedures and make more explorations on other aspects, such as more task combinations, different configurations of MTL procedures, etc. In this way, there is no need to reproduce all MTL procedures that we have spent thousands of GPU hours to run. Other researchers can focus on more advanced explorations, such as zero-shot task grouping.

---

> > ### Comment · Reviewer_QRvu · 2022-08-07
> > **Thank you for the response**
> >
> > I want to thank the authors for clarifications on the questions and am satisfied with the response. I am inclined to keep the score the same and wish the authors the best of luck with their submission.

---

### Official Review · Reviewer_ZASQ · 2022-07-15

**Rating:** 5
**Confidence:** 4
**Soundness:** 2 fair
**Presentation:** 3 good
**Contribution:** 2 fair

**Summary:**

The paper introduces a meta-learning method to learn to predict the transfer gains for each task grouping in a Multi-Task Learning problem. This is based on the assumption that, while the number of task combinations (datapoints in this meta-learning problem) is huge, the data lies in a low-dimensional manifold (manifold hypothesis) and thus can be learned. Since the meta-training set needs to be small (computational cost), the authors also apply active learning to help select the most informative datapoints (groupings) to train.

**Questions:**

See above.

**Limitations:**

Yes

**Strengths And Weaknesses:**

**Strength**
- The idea of the paper is clear. The manifold assumption also seems reasonable, which suggest that we can do better than heuristic estimations of the transfer gain (via learning to predict it).
- The paper obtain good performance when compared with heuristic approaches such as HOA or TAG.

**Weaknesses and Questions**
- One of my main concerns is the computational cost of MTL procedure. The authors mentioned that they consider $k=1$ and $k=(n-1)/2$ so that the number of MTL combinations is roughly equal to TAG and HOA. However, for example, the combinations in HOA only consist of 2 tasks, so might be much less expensive than a combination considered in MTL-Net. Therefore, the evaluation may still be unfair. We should really consider the total computational cost here, instead of the number of task combinations.

- If my understanding is correct, the task embedding (taken out from an embedding table) is independent of the task data. I wonder if we can additionally pass (part of) the task data to a set embedding network to get the task embedding. It might open the door to generalizing to some unseen (but related) tasks at test time.

- Also I found the indicator of transfer gain to be inconsistent in the experiments. For some experiments it is the loss, for some others it is the AUC or the mean absolute error. I am a bit skeptical about this since this might have been cherry-picked. Why don't we use a consistent category across the experiments?

- The authors only present experiments with a hard sharing case of MTL. It is natural to wonder if the method works for soft-sharing of the parameters. Is it the case that we can predict better the transfer gain for hard sharing only?

---

> ### Author Response · Authors · 2022-08-02
> **Response to Reviewer ZASQ (Part 2/2)**
>
> **Response to Question 3**
>
> We appreciate your careful inspections on this paper, and we are confident that there is no risk of “cherry picking”, especially given the extensive experiments on various scenarios and tasks. Instead, the improvements across these diversified tasks validate the generalization capability of our framework.
>
> To be more specific, we evaluate the proposed framework across different applications, including pixel-level image tasks, time-series regression tasks, and clinical classification tasks. These tasks have very different evaluation metrics, and we follow the corresponding evaluation criteria described in the original papers. Thus, we allow the framework to customize the transferring gain as the improvement on a specific metric. In this way, our framework can easily adapt to all kinds of heterogeneous tasks.
>
> **Response to Question 4**
>
> Your question essentially raises an important research direction of the generalizability of multi-task grouping.
>
> As mentioned in HOA [1], when the data or the model architectures changed, the transferring effects among tasks could also distinctly change. We do have similar observations in our experiments. Meanwhile, we also note that there exist certain correlations among the transferring gains obtained from different network architectures. For example, according to the experiments of [1] on the Taskonomy-5 dataset, the correlation coefficient between the transferring gains of a large model and that of a small model can be as large as 0.227. These observations imply that multi-task transferring relationships may be a function of the data, the model, and some other factors, such as optimization algorithms.
>
> Back to your question, when employing a soft-sharing architecture, the underlying transferring relationships are very likely to be different from those based on a hard-sharing MTL architecture. Nonetheless, one important value of this paper lies in establishing the basic framework to capture multi-task transferring relationships given a pre-defined MTL procedure (starting from the hard-sharing architecture in this paper), which can be the foundation to explore the generalization across different configurations of MTL procedures (e.g., different data sets, different model architectures, etc.).
>
> Overall speaking, we value this insightful question very much since it raises a new direction towards more comprehensive and generalizable multi-task grouping.
>
> [1] Standley, Trevor Scott et al. “Which Tasks Should Be Learned Together in Multi-task Learning?” ICML (2020).

---

> ### Author Response · Authors · 2022-08-02
> **Response to Reviewer ZASQ (Part 1/2)**
>
> Thanks very much for your thorough comments. We hope the following specific explanations can help to address your concerns.
>
> **Response to Question 1**
>
> We value your suggestion of including computational cost as a reference and will attach associated information to the revised manuscript. In this work, we prefer to use the number of MTL procedures as a major indicator of computational complexity because this indicator is more intuitive and aligns well with the magnitude of computational cost. The number of tasks in a combination only has marginal effects on the computational cost, which does not affect the major contributions of this paper. We can explicitly mention this point in the revised paper to avoid misunderstanding. Below we would like to give you detailed justifications to address your concern of “unfair evaluation”.
>
>
> Compared with a two-task MTL model, an N-task MTL model (N>2) does have more computational costs due to more task-specific decoders. While in a typical MTL architecture, the encoder part takes up the most computation. For example, on the MIMIC-III-27 benchmark, the computational time of a decoder only takes up 0.3% of the time spent on the encoder part. Thus, the additional decoder-side computations only have marginal effects on the total computational cost. Besides, all these decoders can be computed in parallel. Thus, with proper implementation, the additional computational time due to the introduction of more tasks can be further limited.
>
> Moreover, as shown in Figures 2 & 3(a), you may find that for those $K$s that are significantly less than $(N-1)/2$ (even when $K==1$), we can still obtain remarkable improvements over HOA with largely reduced computational costs. These results explicitly demonstrate the contributions of this paper.
>
> **Response to Question 2**
>
> Yes, your understanding is correct. In this work, we focus on building the structured latent space that can reflect the relationships between task combinations and transferring gains, so we simply employ randomly-initialized task embeddings and optimize these embeddings along with other parameters in MTG-Net solely from meta-training data.
>
> We strongly agree with your comments that it is pretty appealing and invaluable to establish the connection between “task data” (we guess you mean input data and task labels because all tasks share the same data) and task embeddings, since it can endow the meta model with the zero-shot capability to estimate transferring gains for unseen tasks. In the meantime, we have to note that this new research direction faces some critical challenges, such as how to encode an effective task representation from multiple data instances and associated task labels and ensure that such a task representation includes the transferring effects across tasks.
>
> Overall speaking, while this work still focuses on building the fundamental meta learning framework for multi-task grouping, we consider zero-shot task grouping as a truly valuable future direction worthy of greater research attention.

---

> ### Comment · Reviewer_ZASQ · 2022-08-09
> **Thanks for the response**
>
> I would like to thank the authors for the response.
>
> Regarding your point that "Compared with a two-task MTL model, an N-task MTL model (N>2) does have more computational costs due to more task-specific decoders". I would like to remark that this is only true in the case where the task data overlap (almost) completely. In another setting of MTL, where each task has its own dataset (for example, classification and segmentation where each task has a dataset of 224x224 images), running MTL with N tasks is much more expensive than 2 tasks. Note that this setting is also very popular for MTL.
>
> I would suggest the authors to include a discussion about the scope/limitation of the paper: it focuses on the case of hard parameter sharing and overlap of tasks' datasets.

---

> > ### Author Response · Authors · 2022-08-10
> > **Thanks for your suggestion**
> >
> > We very appreciate this suggestion and will take your advice to discuss the limitations of this work in the revised paper. We also agree with you that when each task has its own dataset, the computational cost of N-task MTL will be significantly larger than that of two-task MTL.
> >
> > As you have mentioned, this work focuses on a basic setup of multi-task grouping with a hard-sharing architecture and overlapped task data. Please kindly note that the key motivation of this setup is to ensure a fair and consistent comparison with state-of-the-art task-grouping solutions (e.g., HOA, TAG) and to reveal a critical finding: for a fixed MTL procedure, the transferring relationships induced by task combinations lie in some low-dimensional spaces and thus can be learnable with limited samples.
> >
> > With this new finding, we believe that the proposed framework together with the insightful questions mentioned in your reviews have opened many appealing research opportunities, including but not limited to zero-shot task grouping, generalization across multi-task model architectures, generalization across different datasets (especially non-overlapped task datasets), etc. We will also include a discussion about these potential opportunities.

---

### Official Review · Reviewer_E1on · 2022-07-19

**Rating:** 6
**Confidence:** 3
**Soundness:** 3 good
**Presentation:** 3 good
**Contribution:** 2 fair

**Summary:**

This paper studies the problem of task grouping by using an efficient and effective meta-learning framework. The authors propose to use a meta-learning algorithm, which predicts the per-task performance gains of multi-task learning over single-task learning for any combination, to find optimal task combinations. The key insight is the low-rank hypothesis on task combination space. Based on the hypothesis, the authors develop MTG-Net, which uses an active learning strategy to select meta-learner’s training examples. Experiment results conducted on multiple multitask datasets show significant improvement over state-of-the-art task grouping algorithms.



**Questions:**

Q1  Can a similar hypothesis from ENAS[1]  can be used to further save cost by sharing the parameters of the pair of models for measuring gains?
Q2 How the proposed active learning framework compares with the existing AutoML framework, which models the meta-learner as an RL algorithm?


**Limitations:**

Yes.

**Strengths And Weaknesses:**

Strengths
1. This paper is well written and easy to follow.
2. The approach of introducing a meta-learner to learn task grouping strategies is novel and reasonable.
3. Multiple MTL datasets are used to evaluate the proposed method.

Weaknesses
1. On the efficiency side, even though an active learning algorithm is introduced to reduce the number of samples needed to train the meta-learner, the cost can still be large for training models and measuring the gains. I am wondering if a similar hypothesis from ENAS[1] can be used to further save cost by sharing the parameters of the pair of models for measuring gains.
2.  There exists some related work [2] in AutoML that also leverages the low-rank hypothesis to learn AutoML algorithms based on task-task relationships. Even though the application is slightly different, the novelty of building meta-learned based on low-rank hypothesis is limited.
3.  Even though the author compared it with SOTA for task grouping. I am wondering how the proposed active learning framework compares with the existing AutoML framework, which models the meta-learner as an RL algorithm[1].

[1] Pham, Hieu, et al. "Efficient neural architecture search via parameters sharing." International conference on machine learning. PMLR, 2018.
[2]  Yang, Chengrun, et al. "OBOE: Collaborative filtering for AutoML model selection." Proceedings of the 25th ACM SIGKDD international conference on knowledge discovery & data mining. 2019.

---

> ### Author Response · Authors · 2022-08-02
> **Response to Reviewer E1on (Part 2/2)**
>
> **Response to Weakness 2**
>
> Our work is fundamentally different from OBOE [2] in the literature of AutoML.
>
> The OBOE system aims to speed up the AutoML model selection over an unseen dataset by efficiently collecting the behaviors (e.g., performance, runtime) of a group of models. By assuming that the behaviors of various models have some common patterns across different datasets, given any new dataset, OBOE can only select a subset of models to train and then use its meta model (meta-learned on available datasets) to quickly infer the results of other models. Essentially, the low-rank hypothesis behind OBOE ensures it can generalize the common model behaviors from existing datasets to unseen datasets.
>
> Different from OBOE, our framework focuses on the multi-task grouping problem, in which the critical challenge is to deal with the exponentially growing space of task combinations. More importantly, the low-rank hypothesis used by OBOE does not have the capability to tackle this combinatorial-explosion space. In contrast, our unique finding is that the relationships between task combinations and associated transferring gains lie in some low-dimensional manifolds. This crucial finding motivates the design of our meta learning framework for multi-task grouping.

---

> ### Author Response · Authors · 2022-08-02
> **Response to Reviewer E1on (Part 1/2)**
>
> Thanks very much for your in-depth comments, especially from the perspective of connecting our paper to the related work in NAS/AutoML.
>
>
> **Response to Question 1 & Weakness 1**
>
> Per our understanding, the first question is about how to leverage the parameter-sharing idea in ENAS to speed up the process of measuring MTL transferring gains in our case. (Please kindly correct us if we misunderstand this question.)
>
> While the idea of parameter sharing across MTL procedures may help save computational cost, it could meanwhile take the risk of generating biased   estimations of the transferring gains corresponding to training from scratch. The reason lies in that introducing the parameter sharing across different MTL procedures may bring implicit knowledge transfers among different task combinations, which do not accurately reflect the transferring effects purely from a specific task combination. These biased estimations of transferring gains, being inconsistent with the ground-truth gains obtained by training from scratch, are hence likely to lead to sub-optimal grouping selections.
>
> Therefore, to ensure the essential transferring gains for different task combinations can be accurately estimated, in this paper, we did not leverage the parameter sharing mechanism when performing MTL procedures. While in other scenarios that do not require the accurate results of training from scratch, we may consider increasing the efficiency of MTL procedures via multiple ways, including but not limited to parameter sharing, pre-training & fine-tuning, early stopping, etc.
>
> **Response to Question 2 & Weakness 3**
>
> We really appreciate your proposal of adapting AutoML methods into the multi-task grouping (MTG) problem. Taking ENAS as an example, we can treat all tasks as output nodes and leverage RL to search for a hyper network architecture with separate sub-networks covering specific task combinations. Such an RL-based AutoML view indeed opens a new direction worthy of research for MTG, including a few challenging issues, such as 1) how to designing proper actions to explicitly separate tasks into different groups and assigning informative rewards to these actions; 2) how to design efficient exploration mechanisms to ensure low sample complexity (note that each sample here corresponds to a group of MTL procedures).
>
> Moreover, compared with the RL-based AutoML approach for MTG, our framework has certain distinctive advantages:
>
> - Flexibility.  Our framework consists of two consecutive steps: 1) the estimation of transferring gains and 2) the grouping selection based on these gains. Such a decoupled formulation allows us to adopt more flexible grouping strategies (e.g., imposing constraints on the number of groups or assigning different inference budgets to specific tasks) without re-training the gain estimation model (MTG-Net). Furthermore, we can easily identify the causes of improper groupings, rooted from the inaccurate estimations of gains. This property can help us diagnose the whole procedure and involve human experts in the loop as needed.
> - Interpretability.  As shown in Figure 4 of our main paper, our approach can reveal the inherent manifold of multi-task transferring relationships, which can intuitively inform which group of tasks should be put together and which should be separated from another. Particularly, we can find more insights by further inspecting this structured latent space. For example, as shown in Figure 2 of the appendix, we can identify certain task pairs that consistently lead to positive or negative transfers no matter whether other tasks are involved.
> - Controllable Efficiency. Our framework can offer an explicit tradeoff between efficiency and accuracy via the adjustment of K, which controls N*K rounds of computational-intensive MTL procedures. As shown in Figure 2 of our paper, we demonstrate that even when K = 1, we can still obtain reasonable groupings. Moreover, as Figure 3(a) shows, MTG-Net helps to boost the grouping performance significantly with the increase of K. Owing to the structured latent space of multi-task transferring relationships, a relatively small K (K<= (N-1)/2) can obtain high-quality grouping results that are close to the optimal groupings searched in the space of 2^N task combinations. In contrast, RL-based AutoML approaches require very sophisticated designs of action spaces, exploration mechanisms, and reward schemes to control sample complexity.

---

### Meta-Review · Area_Chair_fNkq · 2022-08-26

**Recommendation:** Accept
**Confidence:** Certain

**Metareview:**

The overall idea of using a meta-learning network with an active learner for grouped multi-task learning is interesting. The experimental results provided in the original submission and rebuttal are extensive to verify the effectiveness of the proposed method. A major limitation of the proposed method is the high computational cost, especially when each task has its own dataset.

Overall, this is a well-written paper that presents an interesting idea for multi-task learning.

**Award:**

No

---

### Decision · Program_Chairs · 2022-09-14

Accept